

# Earthworm-invaded boreal forest soils harbour distinct microbial communities

Justine Lejoly[1*], Sylvie Quideau[1], Jérôme Laganière[2], Justine Karst[1], Christine Martineau[2], Mathew Swallow[3], Charlotte Norris[4], Abdul Samad[2]

[1]Department of Renewable Resources, University of Alberta, Edmonton, Alberta, Canada
[2]Natural Resources Canada, Canadian Forest Service, Laurentian Forestry Centre, Québec City, Québec, Canada
[3]Department of Earth and Environmental Sciences, Mount Royal University, Calgary, Alberta, Canada
[4]Natural Resources Canada, Canadian Forest Service, Pacific Forestry Centre, Victoria, British Columbia, Canada
*Current address: Department of Terrestrial Ecology, Netherlands Institute of Ecology (NIOO-KNAW), Wageningen, the Netherlands

*Correspondence to*: Justine Lejoly (j.lejoly@nioo.knaw.nl)

**Abstract.** Earthworm invasion in North American forests has the potential to greatly impact soil microbial communities by altering soil physicochemical properties, including structure, pH, nutrient availability, and soil organic matter (SOM) dynamics. While most research on the topic has been carried out in northern temperate forests, little is known on the impact of invasive earthworms on soil microbial communities in the boreal forest, a region characterized by a slower decay of organic matter (OM). Earthworm activities can increase OM mineralization, altering nutrient cycling and biological activity in a biome where low carbon (C) and nitrogen (N) availability is typically limiting microbial and plant growth. Here, we characterized and compared microbial communities of earthworm-invaded and non-invaded soils in previously described sites across three major soil types found in the Canadian boreal forest using a space-for-time approach. Microbial communities of forest floors and surface mineral soils were characterized using phospholipid fatty acid (PLFA) analysis and metabarcoding of the 16S rRNA gene, for bacteria and archaea, and ITS2 region for fungi. In forest floors, the effects of earthworm invasion were minor. In mineral soil horizons, earthworm invasion was associated with higher fungal biomass and greater relative abundance of ectomycorrhizal fungi. Oligotrophic bacteria (Acidobacteriota and Chloroflexi) were less abundant in invaded mineral soils, where Gram(+):Gram(-) ratios were also lower, while the opposite was observed for the copiotrophic Bacteroidota. Additionally, earthworm-invaded mineral soils harboured higher fungal and bacterial species diversity and richness. Considering the important role of soil microbial communities for ecosystem functioning, such earthworm-induced shifts in their community composition are likely to impact nutrient cycling, as well as vegetation development and forest productivity at a large scale as the invasion progresses in these boreal systems.



**Graphical abstract**





## Introduction

Microorganisms play a key role in soils, regulating litter decomposition and nutrient and carbon (C) availability. By ingesting litter and mixing it with mineral soil, earthworms can provide microorganisms with a better access to C sources and thus accelerate decomposition and nutrient cycling (Blouin et al., 2013; Curry and Schmidt, 2007; Edwards, 2004). In Canada and the northern United States, the last glaciation extirpated native earthworm populations, resulting in the development of forests devoid of earthworms. These northern forests are now being invaded by exotic earthworm species, which can substantially alter existing SOM dynamics (Frelich et al., 2019). Changes in microbial community composition and diversity resulting from earthworm presence have been observed under laboratory (Chang et al., 2016; Butenschoen et al., 2007; Gómez-Brandón et al., 2010; de Menezes et al., 2018) and field conditions, mostly in temperate forests (Price-Christenson et al., 2020; Dempsey et al., 2013; Groffman et al., 2015; McLean et al., 2006; McLean and Parkinson, 1997).

Comparable studies are minimal for boreal forests despite multiple reports of earthworm invasion occurring throughout North America (Cameron et al., 2007; Moore et al., 2009). In temperate forests, invasive earthworms can be found in more than 80 % of suitable habitats, as reported in an extensive survey of the Great Lakes region of the United States (Fisichelli et al., 2013; Frelich et al., 2019). In boreal forests, 50 % of the low human-impact sites sampled in Alaska by Saltmarsh et al. (2016) were invaded and Cameron and Bayne (2009) estimated that 49 % of the boreal forest of north-eastern Alberta will be invaded by 2059. Impacts of invasive earthworms on soil morphology are similar in temperate and boreal forests and include: (1) thinning of the forest floors and (2) development of novel Ahu surface mineral horizons, characterized by higher pH, clay, and OC content (Bohlen et al., 2004; Lejoly et al., 2021; Lyttle et al., 2011). However, as N limitations for microbial activity are greater in boreal forests (Högberg et al., 2017), the influence of invasive earthworms on microbial communities could be much larger than in temperate forests.

Fungi, including free-living saprotrophs and root-associated symbionts such as mycorrhizal fungi (Lladó et al., 2017), make up a large fraction of biomass in boreal forest soils (Clemmensen et al., 2015). Most boreal tree species form symbiotic associations with ectomycorrhizal (EcM) fungi, except for maple (*Acer* spp.) which associates with arbuscular mycorrhizal (AM) fungi (Brundrett et al., 1990), while *Populus* spp., often categorized as dual-mycorrhizal, are mostly ectomycorrhizal in Western Canada (Karst et al., 2021). Previous studies have reported a positive response of mycorrhizal fungi to earthworm invasion in temperate forests, for both AM (Dempsey et al., 2013; Drouin et al., 2016) and EcM (Jang et al., 2022) fungi. To our knowledge, the response of EcM fungi to earthworm invasion has never been studied in the field for boreal forests (Addison, 2009; Cameron et al., 2012). Because EcM fungi can access C through symbiosis, their growth is mostly limited by N availability, while free-living saprotrophic fungi depend on both soil C and N availability (Högberg et al., 2021, 2017). Saprotrophic fungi are able to decompose a wide range of organic compounds, including lignin and cellulose, with the help of extracellular enzymes (Lebreton et al., 2021). Earthworms, by moving fresh litter down the soil profile (Blouin et al., 2013), could lessen C limitations for saprotrophic fungi and result in their increased relative abundance. However, Crowther et al. (2013) reported that increased grazing by soil fauna (isopods) reduced the abundance of the dominant saprotrophic fungi but





increased fungal diversity as competition with other saprotrophic taxa decreased. With so few field studies, the specific impacts
of earthworm invasion on soil fungal diversity and abundance are not well known.

Similar to fungi, bacteria and archaea decompose plant and soil organic macromolecules, but they are also involved in a wider range of ecosystem processes, including N fixation and transformations (Lladó et al., 2017). While vegetation composition is less important than for fungi (Baldrian, 2017), pH is a key driver of bacterial community composition (Lladó et al., 2017; Fierer and Jackson, 2006). Earthworm gut transit alters bacterial communities both structurally and functionally (Pedersen and
Hendriksen, 1993; Wang et al., 2021; Medina-Sauza et al., 2019). Because of an overall increase in nutrient availability following earthworm invasion, bacteria would be favoured over fungi, the latter being characterized by slower growth and typically decomposing more complex organic compounds (McLean et al., 2006; Soares and Rousk, 2019). In line with this statement, decreased fungi:bacteria ratios have been reported following earthworm invasion in temperate forests (Dempsey et al., 2011), but not systematically (Chang et al., 2017), and this remains to be investigated for boreal forests.

Soil microbial communities are commonly characterized using either DNA metabarcoding or PLFA analysis. These techniques are considered complementary as they show different sensitivities to land use and other environmental changes (Orwin et al. 2018). Both methods have been successfully used to study the impacts of endemic and exotic earthworms on soil microbial communities (Gómez-Brandón et al., 2010; Butenschoen et al., 2007; Chang et al., 2016; Price-Christenson et al., 2020; de Menezes et al., 2018; Dempsey et al., 2013). Although the DNA and PLFA methods used in combination may provide more
robustness to the analysis and conclusions of a study, this approach has been rarely used in past studies assessing earthworm-induced changes on soil bacterial and fungal communities (Shao et al., 2019).

Here, we characterized soil microbial communities of the major boreal forest types of North America, invaded and non-invaded by earthworms, using both metabarcoding and PLFA analysis. Besides the presence or absence of earthworms, our sampling design ensured that other soil forming factors were kept identical, therefore minimizing differences between invaded and non-
invaded areas of the same site. Our objective was to identify shifts in bacterial, archaeal, and fungal communities associated with earthworm invasion, separately for forest floors and mineral soils considering their intrinsically different physicochemical properties and distinct microbial communities (Prescott and Grayston, 2013). We combined two molecular methods (DNA and PLFA analyses) to detect differences in the relative abundance of both functional and taxonomical groups of microorganisms and used PLFA data for quantitative analysis of broad microbial groups. We hypothesized that earthworm
invasion would shift microbial community composition in both mineral soils and forest floors. For fungal communities, we expected no changes in the ratio of ectomycorrhizal to saprotrophic fungi, as earthworm invasion would positively impact them both. Despite differences in soil type and vegetation among sites, we expected shifts to be similar across sites, indicative of increased nutrient availability and pH, including lower fungi:bacteria ratios.



## Materials and methods

**Sample collection**

Sites were selected from the most common soil types and vegetation covers of the Canadian boreal forest, including Brunisols (Soil Classification Working Group, 1998) under sugar maple (*Acer saccharum* Marshall; one site: Valcartier), Podzols under black spruce (*Picea mariana* (Mill.) Britton, Sterns & Poggenburg; one site: Grands Jardins), and Luvisols under trembling aspen (*Populus tremuloides* Mich.; two sites: EMEND and Breton; Fig. 1 & Table S1). The sites used in the present study are
a subset of those described in Lejoly et al. (2021) and correspond to sites with lower anthropogenic impacts. The presence of earthworms was determined by hand-sorting of the litter and hot mustard extraction from the mineral soil surface (Lawrence and Bowers, 2002) and from clear signs of earthworm activity such as the presence of surface casts or extensive bioturbation. Within each site, it was possible to identify earthworm-invaded and earthworm-free patches (Fig. S1). As demonstrated in our previous study, the sampling points were identified to differ only by the presence of earthworms, keeping other soil forming
factors (slope, aspect, vegetation, parent material) as similar as possible (Lejoly et al., 2021). Earthworm presence resulted in thinner forest floors and reworked surface mineral soils (Lejoly et al., 2021), matching previous findings for invaded North American temperate forests (Bohlen et al., 2004; Lyttle et al., 2011; Langmaid, 1964; Alban and Berry, 1994), invaded Fennoscandian arctic forests (Wackett et al., 2018), and typical mull form of humus observed in temperate forests that developed in the presence of earthworms (Muys and Granval, 1997; Beyer et al., 1991).

At each site and for each level of invasion (earthworm-free and earthworm-invaded for EMEND and Grands Jardins; earthworm-free, low-density earthworm-invaded, and high-density earthworm-invaded for Valcartier), three to four samples were collected over the months of June and July 2019 (see Fig. S1). As recommended by Ferlian et al. (2018), we separately sampled the forest floors and the top layers (0-10 cm) of the mineral soils to independently test for the impact of earthworm invasion on these two different soil layers. All samples were stored at -20 ºC for transportation and then at -80 ºC prior to
freeze-drying.



**Figure 1: Map of the different sites selected across the Canadian boreal forest, adapted from Lejoly et al. (2021).**

## DNA extraction and sequencing

Freeze-dried soil samples were ground on a TissueLyser II (Qiagen, Hilden, Germany). Approximately 250 mg of ground

sample were used for DNA extraction using the DNeasy PowerSoil DNA Isolation Kit (Qiagen, Hilden, Germany). Because of its stable nature, the extracted DNA also included non-viable microorganisms (Frostegård et al., 2011). For bacteria and archaea, primers 515F-Y (5'-GTGYCAGCMGCCGCGGTAA-3') and 926R (5'- CCGYCAATTYMTTTRAGTTT-3'; Parada et al., 2016; Rivers, 2016) were used for polymerase chain reaction (PCR) amplification of the V4-V5 regions of the 16S rRNA gene, while for fungi, primers ITS9F (5'-GAACGCAGCRAAIIGYGA-3') and ITS4R (5'- TCCTCCGCTTATTGATATGC-

3'; Rivers, 2016; White et al., 1990) were used for PCR amplification of the internal transcriber spacer 2 (ITS2) region (Integrated DNA Technologies, Coralville, IA, USA). Amplification was carried out following the Platinum SuperFi Green PCR Master Mix protocol (Invitrogen, Carlsbad, CA, USA), with 30 PCR cycles and an annealing temperature of 60 °C for ITS2 and 55 °C for 16S. PCR products were purified using Sera-Mag Select magnetic beads (Cytiva, Marlborough, MA. USA). In preparation for sequencing, a second index PCR was performed with oligonucleotides containing the Illumina specific



overhang adapters. PCR products were again purified as described above. All purified PCR products from both amplified

regions were then pooled using 3–6 µl of each sample to build the amplicon library. DNA concentration and average size were

determined using the Qubit dsDNA HS assay kit on a Qubit fluorometer (Life Technologies, Carlsbad, CA, USA) prior to

running a HS DNA BioAnalyzer (Agilent Technologies, Wilmington, DE, USA) to normalize the pooled library to 4 nM

before sequencing. The pooled libraries were sequenced on an Illumina MiSeq platform using a MiSeq Reagent v3 600 cycles

Kit (Illumina Inc., San Diego, CA, USA) at the University of Alberta Molecular Biological Sciences Unit.

**Bioinformatic analyses**

All bioinformatic analyses were performed in 'Quantitative insights into microbial ecology 2' (QIIME2 version 2021.4

software; Bolyen et al., 2019). Raw sequence data were demultiplexed and quality filtered using the q2-demux plugin followed

by denoising with DADA2 (Callahan et al., 2016). After trimming off the first 20 base pairs (bp) to remove primers, sequence

reads were truncated where the average quality score dropped below 34 (at 285 and 215 bp for forward and reverse 16S reads;

at 260 and 225 bp for forward and reverse ITS reads), and dereplicated with paired end setting to generate amplicon sequence

variants (ASVs) tables containing read counts. Using ASVs over operational taxonomic units results in a finer resolution, to

single nucleotide differences between sequences (Callahan et al., 2017). However, this approach tends to inflate diversity

measurements, 1.3 and 2.1 times higher for fungal and bacterial richness, respectively (Glassman and Martiny, 2018). Mean

amplicon size was $316 \pm 37$ bp ranging from 240 to 433 bp and $372 \pm 2$ bp ranging from 350 to 426 bp for ITS and 16S ASVs,

respectively. After training the naïve Bayes classifiers (q2-feature-classifier), taxonomy was assigned to ASVs using classify-

sklearn (Bokulich et al., 2018) with SILVA 138 SSURef NR99 full-length (99 %) and UNITE 8.3 (97 %) databases for 16S

and ITS2, respectively (Quast et al., 2013; Abarenkov et al., 2010). Only prokaryotic (bacterial and archaeal) or fungal ASVs

were kept and the ASVs tables generated by the QIIME2 were imported into R (version 4.0.5; R Core Team, 2020) for further

analysis. Based on rarefaction curves, a total of 4,437 and 2,832 sequences, for prokaryotic and fungal ASVs respectively,

were randomly selected for analysis of alpha diversity.

For fungal ASVs, functional guilds were determined with FUNGuild database using the FUNGuildR R package (Nguyen et

al., 2016; http://github.com/brendanf/FUNGuildR). Taxa associated with a unique guild ranked probable or highly probable

were selected and grouped into three categories: ectomycorrhiza, pathogens (including parasites), and saprotrophs. Taxa

associated with more than one guild were manually checked and added to the above categories, when suitable, after carefully

reviewing the literature, as recommended by Tedersoo et al. (2022). As general ITS primers are not suitable for AM fungi,

they were not included in functional guild analysis (Stockinger et al., 2010). The proportion of each fungal guild was calculated

as the frequency of reads assigned to a particular guild divided by the total reads counted across all three guilds. Sequence data

were archived at NCBI-SRA (BioProject PRJNA850095).



**Phospholipid fatty acid (PLFA) analysis**

Polar lipids were extracted from freeze-dried samples (3 g of mineral soil and 0.25 to 0.5 g of forest floor) following a modified Bligh and Dyer extraction method (Quideau et al., 2016). Prior to extraction, the PC (19:0/19:0) nonadecanoate surrogate standard (Avanti® Plar Lipids Inc., Alabaster, AL, USA) was added to determine the final recovery. Extracted phospholipids were purified on solid-phase extraction columns (Agilent Technologies, Wilmington, DE, USA) and methylated in mild alkaline environment to obtain fatty acid methyl esters (FAMEs). FAMEs were analyzed on an Agilent 6890 Series capillary GC (Agilent Technologies, Wilmington, DE, USA) equipped with a 25 m Ultra 2 (5 %-phenyl)-methylpolysiloxane column, a flame ionization detector (HewlettPackard, Santa Clara, CA, USA), and He as the carrier gas. PLFA concentrations were determined against the instrument standard methyl decanoate Me10:0 (Aldrich, St. Louis, MO, USA), added prior to GC identification and quantification (Quideau et al., 2016). Results were calculated as a concentration per gram of soil (nmol.g$^{-1}$) using the surrogate standard. PLFA identification was performed using the Sherlock Microbial Identification System version 4.5 software (MIDI, Inc., Newrak, NJ, USA), following standard nomenclature (Maxfield and Evershed, 2014). All unsaturated PLFAs were in cis configuration. Non-microbial PLFAs, with < 14 and > 20 C chain length, were removed before statistical analysis.

Because interpretating specific PLFAs as indicators of specific microbial groups must be done with caution (Frostegård et al., 2011), we only used individual PLFAS widely recognized in the literature as markers of broad microbial groups: 15:0 as general bacteria, i14:0, i15:0, a15:0, i16:0, a16:0, i17:0, and a17:0 for Gram-positive (+) bacteria; 16:1ω7, 18:1ω7, cy17:0, and cy19:0 for Gram-negative(-) bacteria; 18:2ω6 as fungal marker; 20:4ω6, 20:5ω3, and 20:1ω9 for eukaryotes, including fungi, as well as micro- and mesofauna such as protists, nematodes, and microarthropods (Stromberger et al., 2012; Zelles, 1999; Saetre, 1998; Orwin et al., 2018).

**Microbial ratio calculations**

We calculated PLFA ratios corresponding to broad taxonomic groups: the fungi:bacteria ratio as the sum of fungal PLFAs divided by the sum of Gram(+), Gram(-), and general bacterial PLFAs, and the Gram(+):Gram(-) ratio as the sum of Gram(+) PLFAs divided by the sum of Gram(-) bacteria, the latter ratio being negatively correlated with carbon availability (Fanin et al., 2019). In addition, we examined potential changes in PLFAs that have been reported to vary in response to environmental stress as a change in membrane composition for microorganisms (Watzinger, 2015). The cyclo ratio was calculated as the sum of cy19:0ω9 and cy19:0ω7 divided by 18:1ω7, an increase in cyclo-PLFA indicating a response to pH, osmotic, and/or thermal stresses (Yang et al., 2015; Guillot et al., 2000; Mykytczuk et al., 2010). The 10-methyl (10Me) ratio was obtained by dividing 10Me16:0 by 16:0, considering that an increase in 10-methyl branching is linked to an increase in membrane fluidity, which is associated with lower temperatures (Poger et al., 2014).

For bacterial ASVs, the Proteobacteria:Acidobacteriota ratio was calculated as the ratio of the number of reads associated with those two phyla, higher values typically associated with higher nutrient status (Orwin et al., 2018). For fungal ASVs, the



ectomycorrhizal:saprotrophic ratio was also calculated to assess the relative response of these two fungal guilds to earthworm invasion; for instance, this ratio will decrease if earthworm invasion negatively affects EcM fungi and/or positively affects saprotrophic fungi.

**Statistical analyses**

All statistical analyses were performed in R version 4.0.5 (R Core Team, 2021). Threshold for significance was set at alpha = 0.1 to account for the higher probability of type two error associated with the low sample size, recognizing that regional studies such as ours necessarily have low replication. Samples (forest floor and mineral soil) were divided into two categories: invaded and non-invaded, corresponding to the factor "Invasion".

Relative abundances of fungal and bacteria taxa at different taxonomical levels were calculated as the number of reads of the target category divided by the total number of reads. Two indices were calculated for alpha diversity based on rarefied ASV tables: species richness and effective number of species, corresponding to a transformed Shannon diversity index (Jost, 2006). For fungi, those indices were calculated for each functional guild as well as for all fungal ASVs, while for bacteria, they were only based on all ASVs.

To determine whether earthworm invasion affected microbial communities functionally and taxonomically, two-way (factors: invasion and site) analyses of variance (ANOVAs) were performed separately for forest floor and mineral soil samples on fungal and bacterial taxa, diversity indices, and microbial – bacteria and PLFA – ratios presented in the previous section. The ANOVAs were run on bacterial phyla and families present in at least 60 % of the samples and accounting for > 0.5 % of average relative abundance, on fungal phyla, and on fungal guilds (as determined with FAPROTAX) for a functional approach.

fungal guilds while for bacteria, the focus was on phyla, and families. fungal and bacterial phyla, for a taxonomical approach, and on fungal guilds (as determined with FAPROTAX) for a functional approach. When necessary, data were first transformed with Tukey's Ladder of Powers using transformTukey from the rcompanion package to ensure normality of residuals (Package "rcompanion"). Homogeneity of variance was checked with Bartlett test. To account for the unbalanced design for both invasion and site, type II sum of squares was used for the ANOVAs using the car R package (Package "car").

To determine whether earthworm invasion altered soil microbial community composition, permutational analyses of variance (PERMANOVA) were performed on all three datasets (PLFA, 16S, and ITS) after Hellinger transformation using Bray-Curtis distance matrices, for the forest floors and mineral soils separately, with the adonis function from the vegan package (Package "vegan"). Additionally, we ran indicator species analyses on Hellinger transformed data, for the forest floors and mineral soils separately, to identify specific taxa associated with invaded or control samples using the multipatt function from the indicspecies R package (De Caceres and Legendre, 2009).



**Results**

**Earthworm invasion effects on fungi**

A total of 1,735,408 reads were obtained for fungal ASVs, ranging from 70 to 136,741 reads with an average of 34,028 reads per sample. After quality control and trimming, we obtained a total of 10,761 ASVs. On average, 80 % of unique ASVs

belonged to at least one fungal guild. The following numbers of ASVs were identified for each guild: ectomycorrhizal (EcM; 208), saprotrophic (450), and pathogenic (126). Before rarefaction, eight out of 51 samples were removed from the calculations of diversity indices because of low read counts: three invaded forest floors, three invaded mineral soils and two control mineral soils. The relative abundances of fungal phyla, classes, orders, and families are reported in Fig. S2 to S5.

In the forest floors, fungal community composition did not differ by earthworm invasion status (Table 1 & Fig. S6), nor did

the relative abundance of the dominant fungal phyla and guilds (Fig. 2 & Tables S2 & S3). However, the indicator species analysis revealed that the EcM genus *Lactarius* was associated with control forest floors (Table 2). Species richness and diversity were not affected by earthworm invasion in forest floors (Table 3). In the mineral soils, fungal community composition differed by earthworm invasion status (Table 1 & Fig. S6). The relative abundance of Ascomycota, the most abundant fungal phylum, was significantly lower in earthworm-invaded mineral soils (Table S2). The relative abundance of

dominant fungal guilds was also affected by earthworm invasion, with lower relative abundance of saprotrophic fungi (40 % compared to 60% in control soils) and higher relative abundance of EcM fungi (58 % compared to 35 % in control soils; Fig. 2 & Table S3). As a result, the fungal communities shifted from saprotroph- to EcM-dominated and the ectomycorrhizal:saprotrophic ratio increased from 1.25 to 2.42 (Table 4). However, diversity and species richness remained higher for saprotrophic fungi (16 and 30, respectively) than for EcM fungi (7 and 17, respectively).

The indicator species analysis identified the EcM genera *Amphinema* and *Tomentella* as indicators of invasion in mineral soils (Table 2). Fungal species richness and diversity were higher in invaded mineral soils for EcM (+104 and 53 %, respectively), saprotrophic (+71 and 61 %, respectively), and pathogenic (+150 and 145 %, respectively) fungi, while only fungal species richness was higher when considering all fungi regardless of their functional guild (Table 3 & S3).



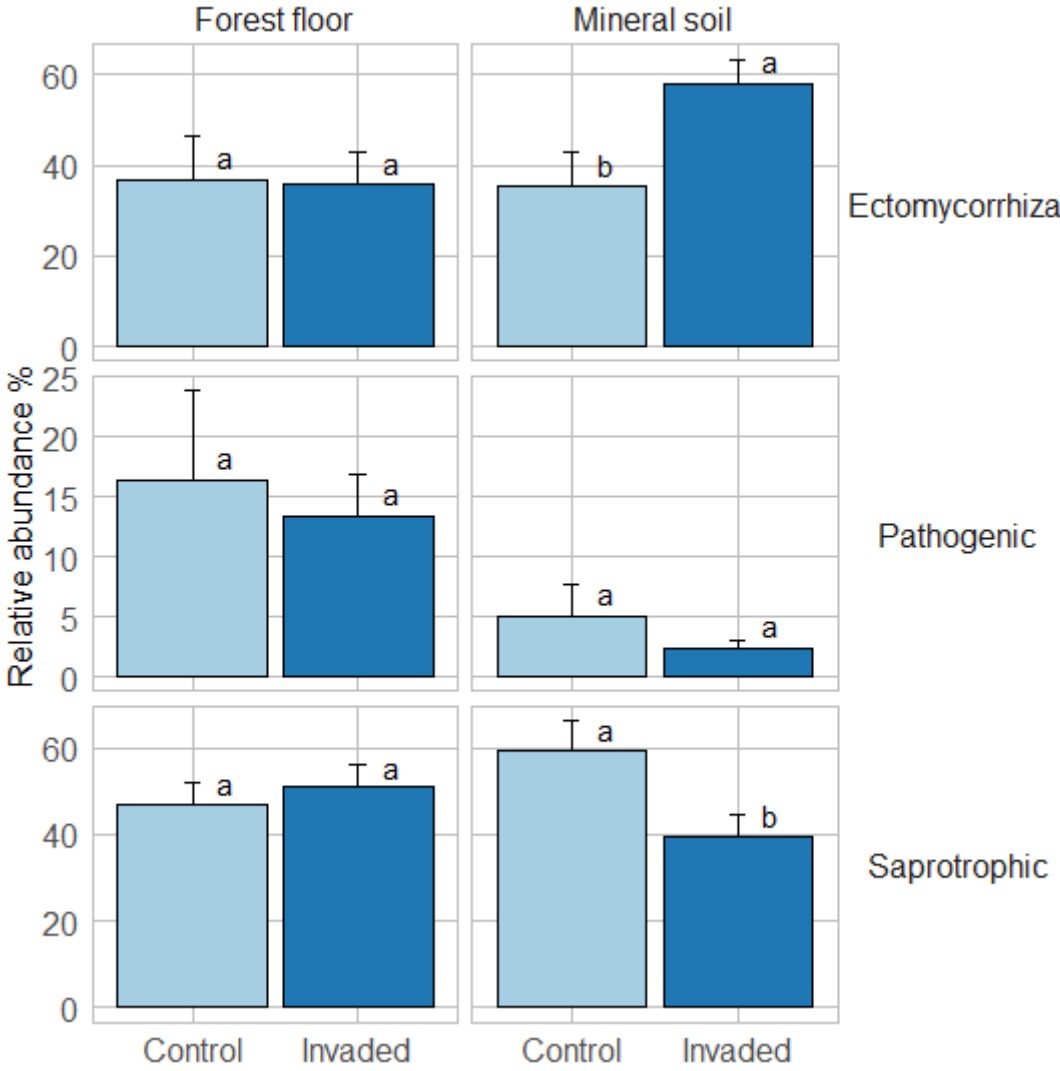

**Figure 2: Average (±1 SE) relative abundances (as number of reads) of fungal guilds. Different letters indicate significant differences (p-value < 0.1) between control and earthworm invaded forest floors or mineral soils and were obtained with the posthoc HSD Tukey test after two-way analysis of variance (n=7–18).**





**Table 1: Permutational analysis of variance performed on Hellinger-transformed data for fungal and bacterial amplicon sequence variants and phospholipid fatty acids (PLFA) using Bray-Curtis distance matrix for earthworm invaded (EW) and non-invaded (CONT) forest floors (LFH) and mineral soils (MIN). Df stands for degrees of freedom and Sum sq for sum of squares.**

| | | | Df | Sum sq | F | $R^2$ | p-value |
|---|---|---|---|---|---|---|---|
| Fungi | Forest floor | Invasion | 1 | 0.13 | 0.94 | 0.03 | 0.49 |
| | | Site | 2 | 2.08 | 7.60 | 0.44 | <0.001 |
| | | Interaction | 2 | 0.23 | 0.83 | 0.05 | 0.66 |
| | | Residuals | 17 | 2.32 | | 0.49 | |
| | | Total | 22 | 4.76 | | 1.00 | |
| | Mineral soil | Invasion | 1 | 0.33 | 2.22 | 0.06 | <0.01 |
| | | Site | 2 | 1.51 | 5.13 | 0.28 | <0.001 |
| | | Interaction | 2 | 0.40 | 1.35 | 0.07 | 0.07 |
| | | Residuals | 21 | 3.09 | | 0.58 | |
| | | Total | 26 | 5.33 | | 1.00 | |
| Bacteria | Forest floor | Invasion | 1 | 0.15 | 1.79 | 0.04 | 0.09 |
| | | Site | 2 | 2.02 | 12.49 | 0.53 | <0.001 |
| | | Interaction | 2 | 0.17 | 1.03 | 0.04 | 0.37 |
| | | Residuals | 18 | 1.46 | | 0.38 | |
| | | Total | 23 | 3.80 | | 1.00 | |
| | Mineral soil | Invasion | 1 | 0.20 | 2.90 | 0.07 | <0.01 |
| | | Site | 2 | 1.14 | 8.20 | 0.38 | <0.001 |
| | | Interaction | 2 | 0.21 | 1.49 | 0.07 | 0.11 |
| | | Residuals | 21 | 1.46 | | 0.49 | |
| | | Total | 26 | 3.01 | | 1.00 | |
| PLFA | Forest floor | Invasion | 1 | 0.01 | 0.45 | 0.01 | 0.74 |
| | | Site | 2 | 0.21 | 9.83 | 0.50 | <0.001 |
| | | Interaction | 2 | 0.01 | 0.48 | 0.02 | 0.85 |
| | | Residuals | 18 | 0.19 | | 0.46 | |
| | | Total | 23 | 0.41 | | 1.00 | |
| | Mineral soil | Invasion | 1 | 0.01 | 1.87 | 0.04 | 0.08 |
| | | Site | 2 | 0.12 | 10.34 | 0.44 | <0.001 |
| | | Interaction | 2 | 0.02 | 1.37 | 0.06 | 0.18 |
| | | Residuals | 22 | 0.13 | | 0.47 | |
| | | Total | 27 | 0.27 | | 1.00 | |





**Table 2: Indicator species for ectomycorrhizal and saprotrophic fungi, as well as bacteria, associated with control or earthworm invaded forest floors and mineral soils. Indicator species were obtained using the multipatt function in the indicspecies R package on Hellinger-transformed data. The amplicon sequence variants (ASVs) were grouped at the finest identified taxonomic level between order and genus, taxa only identified to the class level were excluded. The statistic is a combination of specificity (A=1 when only found in the one group) and fidelity (B=1 when found in all samples of the one group). Relevant indicators were selected with B > 0.5 and relative abundance > 0.05 % in the taxonomical group for which they were selected as indicator. No indicator species were found for pathogenic fungi.**

|  |  |  | Invasion | Taxonomical group | A | B | Stat | p-value |
|---|---|---|---|---|---|---|---|---|
| **Fungi** | Ectomycorrhizal | Forest floor | Control | *Lactarius* | 0.88 | 0.60 | 0.73 | 0.03 |
|  |  | Mineral soil | Invaded | *Tomentella* | 0.94 | 0.71 | 0.82 | 0.010 |
|  |  |  |  | *Amphinema* | 0.95 | 0.65 | 0.79 | 0.010 |
|  | Saprotrophic | Forest floor | Control | *Syzygospora* | 0.67 | 1.00 | 0.82 | 0.013 |
|  |  |  |  | *Umbelopsis* | 0.63 | 1.00 | 0.79 | 0.026 |
| **Bacteria** |  | Forest floor | Control | AD3 | 0.75 | 0.57 | 0.66 | 0.048 |
|  |  |  | Invaded | *Parafilimonas* | 1.00 | 0.59 | 0.77 | 0.019 |
|  |  |  |  | *Rhodococcus* | 0.88 | 0.65 | 0.76 | 0.038 |
|  |  | Mineral soil | Control | Unidentified Caulobacteraceae | 0.72 | 0.78 | 0.75 | 0.019 |
|  |  |  | Invaded | Uncultured BD7 | 0.94 | 0.67 | 0.79 | 0.005 |
|  |  |  |  | *Haliangium* | 0.89 | 0.67 | 0.77 | 0.010 |
|  |  |  |  | *Pirellula* | 0.83 | 0.72 | 0.77 | 0.014 |
|  |  |  |  | *Fimbriiglobus* | 0.88 | 0.67 | 0.77 | 0.009 |
|  |  |  |  | Uncultured OM190 | 0.82 | 0.61 | 0.71 | 0.039 |
|  |  |  |  | *Anaeromyxobacter* | 1.00 | 0.50 | 0.71 | 0.028 |
|  |  |  |  | Mle1 | 1.00 | 0.50 | 0.71 | 0.026 |

**Table 3: Averages (±1 SE) of fungal and bacterial species richness and diversity (as the effective number of species) indices (n=7–15). Different letters represent significant differences between control and earthworm invaded soils (p-value < 0.1) and are presented separately for the forest floors and mineral soils.**

|  |  |  | Forest floor | | Mineral soil | |
|---|---|---|---|---|---|---|
|  |  |  | Control | Invaded | Control | Invaded |
| **Fungi** | Global |  |  |  |  |  |
|  |  | Richness | 155 (10) a | 143 (10) a | 63 (7) b | 102 (8) a |
|  |  | Diversity | 42 (6) a | 37 (4) a | 18 (3) a | 28 (5) a |
|  | Ectomycorrhizal |  |  |  |  |  |
|  |  | Richness | 14 (4) a | 10 (3) a | 7 (2) b | 12 (1) a |
|  |  | Diversity | 5 (2) a | 5 (1) a | 4 (1) a | 5 (1) a |
|  | Saprotrophic |  |  |  |  |  |
|  |  | Richness | 34 (4) a | 31 (3) a | 16 (2) b | 24 (3) a |
|  |  | Diversity | 14 (2) a | 13 (2) a | 7 (1) b | 11 (1) a |
|  | Pathogenic |  |  |  |  |  |
|  |  | Richness | 7 (2) a | 6 (1) a | 1 (1) b | 3 (1) a |
|  |  | Diversity | 3 (1) a | 3 (1) a | 1 (1) b | 2 (1) a |
| **Bacteria** |  | Richness | 216 (32) b | 270 (22) a | 179 (12) b | 217 (8) a |
|  |  | Diversity | 94 (19) a | 114 (13) a | 65 (6) b | 81 (4) a |





**Table 4: Averages (±1 SE) of selected fungal (n=7–18), bacterial (n=7–18), and phospholipid fatty acid (PLFA; n=8–19) ratios. Different letters represent significant differences between control and earthworm invaded soils (p-value < 0.1) and are presented separately for the forest floors and mineral soils. The 10Me ratio was obtained by dividing the PLFA 10Me16:0 by 16:0; the Cyclo ratio was calculated by dividing the sum of cy19:0ω9 and cy19:0ω7 by 18:1ω7.**

| | Forest floor | | Mineral soil | |
|---|---|---|---|---|
| | Control | Invaded | Control | Invaded |
| **Fungi** | | | | |
| Ectomycorrhizal:Saprotrophic | 0.94 (0.31) a | 1.48 (0.63) a | 1.25 (0.78) b | 2.42 (0.54) a |
| **Bacteria** | | | | |
| Proteobacteria:Acidobacteriota | 2.15 (0.21) a | 2.53 (0.31) a | 0.64 (0.05) b | 0.86 (0.07) a |
| **PLFA** | | | | |
| Fungi:Bacteria | 0.41 (0.06) a | 0.49 (0.08) a | 0.09 (0.03) b | 0.11 (0.03) a |
| Gram(+):Gram(-) | 0.55 (0.05) a | 0.60 (0.04) a | 0.72 (0.02) a | 0.66 (0.02) b |
| 10Me ratio | 0.18 (0.05) a | 0.18 (0.04) a | 0.67 (0.05) a | 0.56 (0.04) b |
| Cyclo ratio | 0.45 (0.10) a | 0.39 (0.05) a | 0.52 (0.07) a | 0.42 (0.03) b |

**Earthworm invasion effects on bacteria and archaea**

For bacterial ASVs, a total of 2,302,500 reads was obtained with an average of 45,147 per sample, ranging from 35,530 to 54,272 reads. After quality control and trimming, we obtained a total of 17,086 ASVs. While the term 'bacterial communities' is used here, archaea were not excluded. The relative abundances of bacterial phyla, classes, orders, and families can be found in Fig. S7 to S10.

In the forest floors, bacterial community composition did not differ by earthworm invasion status (Table 1 & Fig. S11). However, specific taxonomic groups were affected, with the relative abundance of Actinobacteriota significantly lower in invaded (24 %) compared to the control soils (32 %; Fig. 3 and Table S4). Verrucomicrobiota increased significantly in earthworm-invaded forest floors, reaching 3 % of relative abundance, mainly due to the increase in the Chthoniobacteraceae family. Within the Bacteroidota phylum, the Chitinophagaceae family was also positively affected by earthworm invasion, with the genus *Parafilimonas* identified as an indicator of invasion in the forest floor (Table 2).

In the mineral soils, earthworm invasion altered the composition of bacterial communities, according to the PERMANOVA results (Table 1 & Fig. S11). Notably, the relative abundance of Acidobacteriota significantly decreased from 27 % to 21 % following earthworm invasion, resulting in a higher Proteobacteria:Acidobacteriota ratio (Fig. 3 & Table 4). While Actinobacteriota were not affected by earthworm invasion at the phylum level, the Gaiellaceae family responded positively. The Chitinophagaceae family was also positively affected by earthworm invasion in the mineral soil, mirroring the significant increase in Bacteroidota. Although their relative abundance was much lower (< 10 %), it is still worth noting that the Gemmatimonadota were positively affected by earthworm invasion in the mineral soil. Within the Chloroflexi, the uncultured bacteria 1921-2, 1921-3, and B12WMSP1 were indicators of non-invaded mineral soils. The Hyphomicrobiaceae and Nitrosomonadaceae families were positively affected by earthworm invasion. Similarly, the relative abundance of Pirellulaceae was significantly higher in earthworm-invaded mineral soils, of which *Pirellula* was selected as indicator together with two



other Planctomycetota: *Fimbriiglobus* and the uncultured bacterium OM190 (Table 2). Lastly, these shifts were associated

with higher species richness and diversity in invaded mineral soils (Table 3).

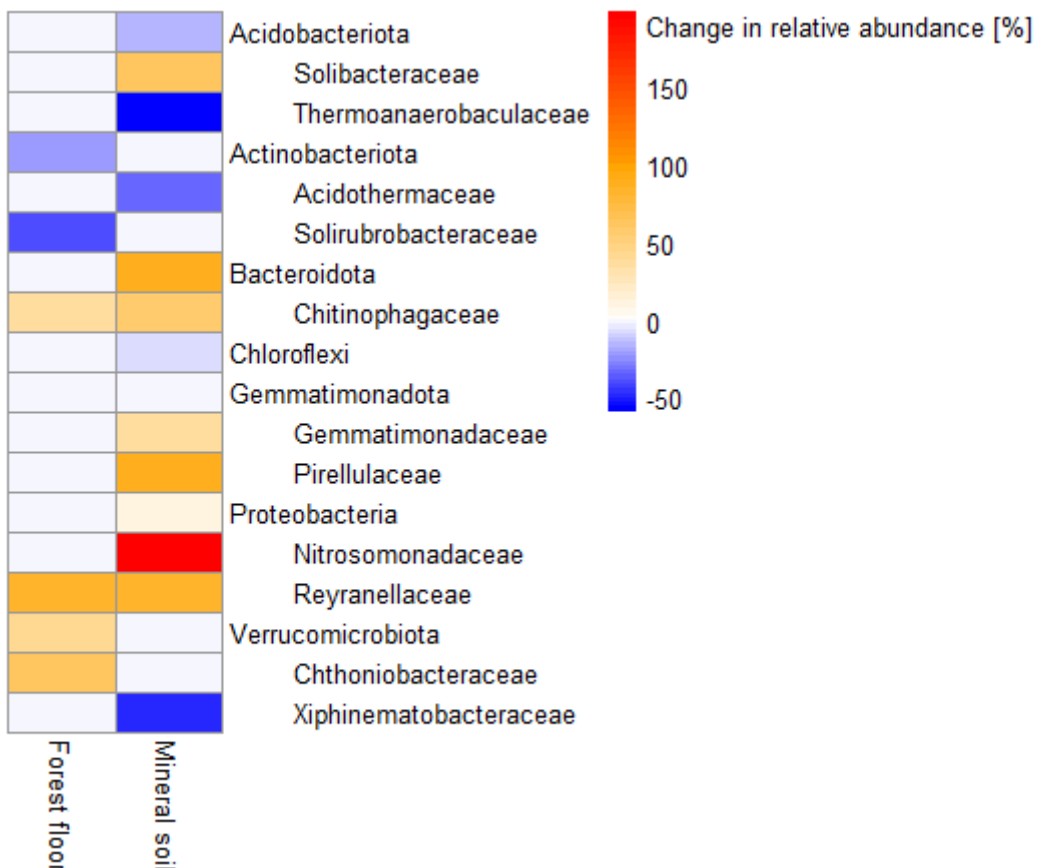

**Figure 3: Heatmap of changes in relative abundance (as number of reads) of bacterial phyla and families. Only significant differences**
**(p-value < 0.1), determined by two-way analyses of variance (ANOVAs) followed by posthoc HSD Tukey test, are displayed (n=7–**
**18). Orange/red hues represent an increase and blue hues, a decrease. Results of the two-way ANOVAs associated with the significant**
**differences can be found in Table S3.**

**Earthworm invasion effects on microbial and faunal PLFAs**

Results of the permutational analysis of variance (PERMANOVA) showed that invasion did not significantly affect the overall

PLFA composition of the forest floors and mineral soils (Table 1 & Fig. S12). While the total PLFA biomass was not affected

by earthworm invasion, the fungal PLFA 18:2ω6 was significantly higher in earthworm-invaded mineral soils, with a

concentration almost twice as high as in the control soils of the same soil layer (Table 5 & Fig. 4). Consequently, the

fungi:bacteria ratios were slightly but significantly higher in earthworm-invaded mineral soils (0.11) compared to the controls

(0.09), as the concentrations of bacterial PLFAs were not affected by earthworm invasion. Additionally, the concentration of





eukaryotic PLFAs (20:4ω6, 20:5ω3, and 20:1ω9) was higher in earthworm-invaded mineral soils. Also in the mineral soils, the Gram(+):Gram(-) ratio was significantly lower after earthworm invasion, as were the 10Me and cyclo ratios, indicative of environmental stress (Tables 4 & S5). In earthworm-invaded forest floors, the observed changes are limited to higher concentrations of Gram(+) bacteria PLFA i14:0 and a17:0, and of the eukaryotic PLFA 20:4ω6, while PLFA ratios are not affected by earthworm invasion. For all PLFA concentrations and ratios, the interaction between invasion and site was non-310  significant (Table S5).

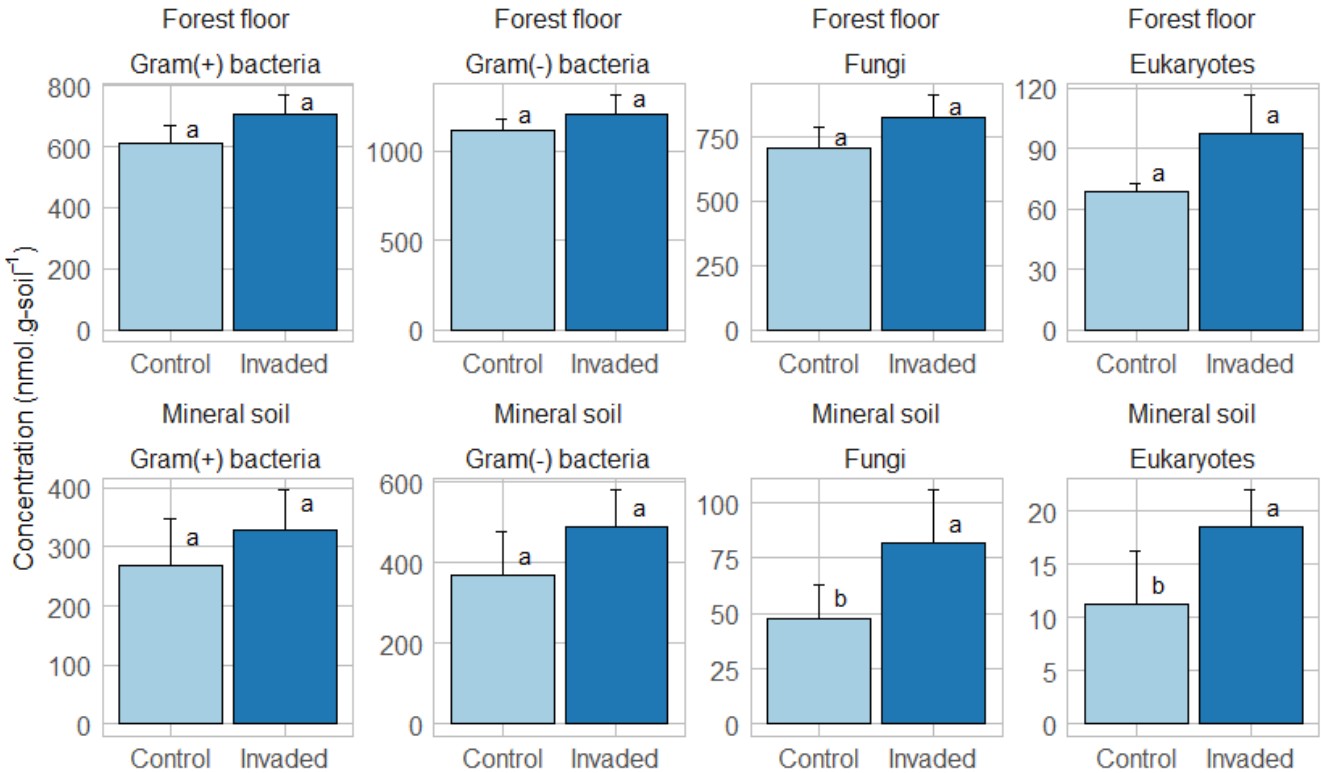

**Figure 4: Average (±1 SE) concentrations (nmol.g-soil-1) of major phospholipid fatty acid groups. Different letters indicate significant differences (p-value < 0.1) between control and earthworm invaded forest floors or mineral soils and were obtained with the posthoc HSD Tukey test after two-way analysis of variance (n=8–19).**





**Table 5:** Average (±1 SE) concentrations (nmol.g-soil-1) of fungal phospholipid fatty acids (PLFAs) associated with specific microbial (general bacteria, Gram(+), Gram(-), and fungi) and eukaryotic, including micro- and mesofauna, groups (n=8–19). Different letters represent significant differences between control and earthworm invaded soils (p-value < 0.1) and are presented separately for the forest floors and mineral soils.

| | Forest floor | | Mineral soil | |
|---|---|---|---|---|
| | Control | Invaded | Control | Invaded |
| Total PLFA | 4909 (277) a | 5760 (357) a | 1317 (351) a | 1769 (346) a |
| General bacteria (15:0) | 47 (4) a | 54 (6) a | 10 (2) a | 14 (4) a |
| Gram(+) bacteria | 613 (55) a | 705 (64) a | 269 (78) a | 328 (68) a |
| i14:0 | 29 (6) b | 34 (3) a | 6 (2) a | 9 (3) a |
| i15:0 | 263 (31) a | 290 (34) a | 108 (31) a | 130 (25) a |
| a15:0 | 133 (16) a | 156 (11) a | 72 (28) a | 78 (18) a |
| i16:0 | 89 (11) a | 106 (12) a | 43 (11) a | 53 (13) a |
| a16:0 | 11 (2) a | 12 (2) a | 2 (1) a | 4 (1) a |
| i17:0 | 35 (4) a | 39 (5) a | 20 (5) a | 26 (5) a |
| a17:0 | 52 (7) b | 67 (9) a | 18 (5) a | 27 (6) a |
| Gram(-) bacteria | 1114 (67) a | 1215 (105) a | 370 (106) a | 489 (91) a |
| 16:1ω7 | 364 (20) a | 429 (31) a | 87 (22) a | 130 (29) a |
| 18:1ω7 | 457 (57) a | 464 (63) a | 161 (50) a | 217 (40) a |
| cy17:0 | 119 (8) a | 132 (9) a | 37 (13) a | 51 (13) a |
| cy19:0 | 6 (4) a | 5 (3) a | 5 (4) a | 1 (1) a |
| Fungi (18:2ω6) | 707 (83) a | 828 (84) a | 48 (16) b | 82 (25) a |
| Eukaryotes | 69 (4) a | 97 (19) a | 11 (5) b | 19 (4) a |
| 20:4ω6 | 30 (3) b | 44 (8) a | 4 (3) b | 8 (2) a |
| 20:5ω3 | 12 (2) a | 26 (13) a | 3 (2) a | 4 (1) a |
| 20:1ω9 | 26 (4) a | 27 (5) a | 4 (2) a | 6 (1) a |

## Discussion

### Earthworm invasion and shifts in microbial community composition

The effects of earthworm invasion on soil microbial communities were similar across sites spanning the Canadian boreal forest, regardless of soil type. Although the factor 'site' was in most cases significant, the interaction between invasion and site was non-significant, indicating that despite potentially variable amplitudes, earthworm invasion has similar effects on the most common mineral soil types found in the boreal forest biome.

Our results corroborate previously reported changes in bacterial communities associated with earthworm activity. De Menezes et al. (2018) reported an increase in Verrucomicrobiota and Gong et al. (2018) an increase in Gemmatimonadota after incubation with earthworms in pasture and arable soils, respectively. Chitinophagaceae, involved in the degradation of chitin and other complex sugars such as cellulose and hemicelluloses (Rosenberg, 2014), were also previously positively correlated with earthworm presence (de Menezes et al., 2018). The observed increase in Proteobacteria:Acidobateriota ratios, indicative of higher nutrient status (Smit et al., 2001), has been previously associated with earthworm activity in arable soils (Gong et al., 2018).



While we initially hypothesized that the fungi:bacteria ratios would decrease following invasion, we observed the opposite, and measured an overall increase in fungal biomass in mineral soils, estimated by the fungal PLFA 18:2ω6. This should be added to the list of contradicting results previously documented. Dempsey et al. (2011) reported a decrease in fungi:bacteria ratios in a temperate hardwood forest on an area basis (m$^2$), but not per gram of soil, which is how we calculated our ratio. Similarly, in an incubation with arable soil, fungal biomass decreased in the presence of earthworms (Butenschoen et al., 2007). Chang et al. (2017) observed an increase in fungi:bacteria ratios, but only in the presence of multiple earthworm species in deciduous temperate forests, consistent with our findings considering that most of our sites were also invaded by multiple earthworm species. Similarly, in a Chinese subtropical forest, the presence of the exotic earthworm *Pontoscolex corethrurus*, native to South America, also increased fungi:bacteria ratios (Shao et al., 2019).

The analysis of fungal guilds revealed that the positive impact of earthworm invasion on fungal biomass was limited to EcM fungi present in the mineral horizons of the studied boreal forest soils. Previous field studies observed a similar response for EcM and AM fungi following earthworm invasion in temperate forests (Dempsey et al., 2013; Drouin et al., 2016; Jang et al., 2022). In comparison, in a mesocosm experiment, Cameron et al. (2012) reported that the presence or density of invasive earthworms did not affect EcM fungal community composition or colonization of white spruce roots. However, only a subset of EcM fungi able to withstand disturbance was likely present in the pots used in this mesocosm study and the seedlings were grown in conditions not conducive to growth of fine roots, a required process for changes in EcM fungal colonization to occur. Thus, the conditions of Cameron et al. (2012) may have underestimated the effects of earthworms. The observed decrease in the relative abundance of saprotrophic fungi goes against our initial hypothesis. Saprotrophic fungi and bacteria both decompose OM compounds, but bacteria are typically more competitive in degrading simple C compounds (Rousk and Bååth, 2011). An increase in those simple C compounds in the presence of earthworms would explain the observed decrease in the relative abundance of saprotrophic fungi, giving bacteria a competitive advantage. However, as the fungal biomass (estimated by the fungal PLFA 18:2ω6) increased, this might not correspond to an absolute decrease in saprotrophs.

Similar to fungi:bacteria ratios, there is also mixed responses to earthworm presence in Gram(+):Gram(-) ratios. In line with our findings, Butenschoen et al. (2007) found that endogeic earthworms favoured Gram(-) over Gram(+) bacteria in arable soils. On the other hand, Dempsey et al. (2013) saw a decrease in Gram(-) and an increase in Gram(+) bacteria in the surface mineral soils after earthworm invasion in a northern hardwood forest. Other studies found variable responses of the relative abundance of Gram(-) and Gram(+) bacteria among earthworm species (Chang et al., 2016, 2017). Taken together, the absence of consensus among studies about the impact of earthworms on the fungi:bacteria and Gram(+):Gram(-) ratios shows that the relationships are complex and not well understood. There is however evidence that pH and C availability, among other factors, can alter the relative abundance of those broad microbial groups (Rousk et al., 2010; Fanin et al., 2019; Zhou et al., 2017).

**Microbial community composition and potential environmental changes**

The current invasion of North American forests by exotic earthworms can have cascading effects on ecosystem functioning (Frelich et al., 2019). Although microbial communities in forest floors were relatively insensitive, this soil layer often



disappears with earthworm invasion (Lejoly et al., 2021). In contrast, the more pronounced changes occurred in the mineral soil, whose relative relevance to soil nutrient cycling will increase because of the loss of forest floor. Given that soil microbial communities play a central role in nutrient cycling and SOM dynamics (Simpson et al., 2007), the shifts observed in the mineral soil are likely to have far-reaching consequences for nutrient availability and ecosystem functioning.

Although less pronounced than in the mineral soils, taxonomic shifts in the forest floors match the lower TOC and TN content previously observed at these sites (Table S1; Lejoly et al., 2021). Notably, the EcM fungal genus *Lactarius,* identified as indicator of absence of earthworms in the forest floor, is often classified as nitrophilic (Lilleskov et al., 2002, 2011). Rodriguez-Ramos et al. (2020) also found the same taxon to be negatively affected by wildfire and salvage-logging, which both alter the integrity of the forest floor, as does earthworm invasion. Verrucomicrobiota, including the Chthoniobacteraceae family,

increased in invaded forest floors and are often considered oligotrophic (Orwin et al., 2018; Hu et al., 2022; Sun et al., 2017; Ramirez et al., 2012). In comparison, the relative abundance of Actinobacteriota, classified as copiotrophs (Ramirez et al., 2012; Orwin et al., 2018), decreased in invaded forest floors. This is in line with the common view that, by feeding on more palatable substrates, earthworms lead to the accumulation of more recalcitrant C compounds in forest floors (Curry and Schmidt, 2007).

Numerous studies have previously shown that earthworms increase C availability, through the incorporation of fresh litter into the soil, priming microbial communities with labile C and thus enhancing the decomposition of more recalcitrant OM (Bohlen et al., 2002; Medina-Sauza et al., 2019; De Graaff et al., 2010; Fontaine et al., 2004). Our results corroborate these observations, with lower Gram(+):Gram(-) ratios in earthworm-invaded mineral soils, indicating higher C availability (Fanin et al., 2019). Additionally, the lower relative abundance of the oligotrophic Acidobacteriota and Chloroflexi and the higher relative

abundance of the copiotrophic Bacteroidota further illustrate that the microbial community is more adapted to higher C availability in earthworm-invaded mineral soils (Fierer et al., 2007; Zeng et al., 2022). Bacteroidota, as well as Chitinophagaceae, are also often positively correlated with high N additions in boreal forests (Högberg et al., 2014), which aligns with the numerically higher TN content of invaded mineral soils (Table S1). Additionally, the EcM fungal genera *Aphimena* and *Tomentella*, indicator of earthworm invasion in the mineral soil, have been positively correlated with soil

fertility and pH in boreal forests (Sterkenburg et al., 2015; Haas et al., 2018). While we did not measure nutrient availability, the differences in microbial community composition of earthworm-invaded and control mineral soils might indicate a shift in C and N cycling associated with earthworm invasion.

In the mineral soils, the response of the PLFA ratios suggests that earthworm invasion affected environmental conditions for microbial activity. The decrease observed for the cyclo ratio (i.e. the sum of PLFAs cy19:0ω9 and cy19:0ω7 divided by

18:1ω7) has been previously associated with a decrease in environmental stress of osmotic or acidic origin (Yang et al., 2015; Guillot et al., 2000; Mykytczuk et al., 2010), which is consistent with the higher pH found in earthworm-invaded mineral soils (Table S1; Lejoly et al., 2021). The increase in pH can also explain the higher bacterial diversity and richness (Fierer and Jackson, 2006) and the lower Gram(+):Gram(-) ratios (Frostegård et al., 1993) associated with earthworm-invaded mineral soils. In addition, the lower 10Me ratios observed in invaded mineral soils could indicate a decrease in membrane fluidity,



linked to higher temperatures (Poger et al., 2014; Kieft et al., 1994; Zhang and Rock, 2008). The thinning of the forest floor layer observed in earthworm-invaded soils (Lejoly et al., 2021) could increase soil temperature during the growing season, similarly to the mechanical removal of forest floor in managed forests (Tan et al., 2005).

While some of our observations align with previous findings for earthworm invasion in temperate forests, there are some interesting divergences, including the relative overall increase in fungi compared to bacteria and the decrease in

Gram(+):Gram(-) ratios in invaded mineral soils. In boreal forests, fungi:bacteria ratios are typically higher than in temperate forests, as fungi have slower growth and lower nutrient requirements compared to bacteria (Soares and Rousk, 2019) and fungal communities therefore play a major role in C sequestration (Francisco et al., 2016; Clemmensen et al., 2013; Chen et al., 2020). As EcM fungi can slow down C cycling (Averill and Hawkes, 2016), their increase in invaded mineral soils could positively affect soil C storage in boreal forests, which contain more than twice as much SOC as temperate forests per unit

area (Lal, 2005). However, because AM fungi were excluded from this study, we cannot conclusively state that fungal communities became EcM-dominated in all invaded soils. Considering the inherent differences between the two forest ecosystems, extrapolating the findings of temperate forests to boreal forests is therefore not recommended. Our results suggest that boreal forests are affected by invasive earthworms differently from temperate forests and confirm the importance to implement additional studies in the boreal biome, including enzyme analyses and/or RNA techniques to target the activity of

microbial communities.

**Conclusions**

Using a combined PLFA and metabarcoding approach, our study presented the first evidence of microbial community composition shifts associated with earthworm invasion in boreal forest soils. Both fungal and bacterial communities were affected, but changes differed between forest floors and mineral soils, with more pronounced shifts in the latter. This result is

important because as the thickness and C stocks of forest floors drastically decrease with earthworm invasion (Lejoly et al., 2021), the relative importance of forest floor microbial communities for soil nutrient cycling will likely decrease. In contrast, the microbial communities of earthworm-invaded mineral soils, whose C stocks are not affected by invasion (Lejoly et al., 2021), were quite distinct from that of pre-invasion mineral soils. Earthworm-invaded mineral soils harboured greater fungal biomass, associated with higher relative abundance of EcM fungi, and higher microbial diversity. Shifts in bacterial

communities included lower Gram(+):Gram(-) ratios, higher relative abundance of copiotrophic phyla, and lower relative abundance of oligotrophic phyla, all indicating higher nutrient and C availability in earthworm-invaded soils. The similar shifts observed across three intrinsically different soil types of North American boreal forests constitute strong evidence that earthworm invasion has universally applicable effects on soil microbial community composition in these ecosystems.



**Code availability**

The R and QIIME2 codes used for (bio)statistical analyses are available upon request.

**Data availability**

The sequencing data were deposited on the NCBI Sequence Reach Archive under the BioProject PRJNA850095. The PLFA and other supporting data will be made available on DataDryad upon manuscript acceptance.

**Author contributions**

Justine Lejoly: Conceptualization, Data curation, Formal analysis, Funding acquisition, Investigation, Methodology, Project administration, Visualization, Writing – original draft preparation. Sylvie Quideau: Conceptualization, Funding acquisition, Methodology, Resources, Supervision, Writing – review & editing. Jérôme Laganière: Conceptualization, Funding acquisition, Methodology, Writing – review & editing. Justine Karst: Conceptualization, Resources, Writing – review & editing. Christine Martineau: Conceptualization, Funding acquisition, Resources, Writing – review & editing. Mathew Swallow:
Conceptualization, Writing – review & editing. Charlotte Norris: Conceptualization, Writing – review & editing. Abdul Samad: Software, Writing – review & editing.

**Competing interests**

The authors declare that they have no conflict of interest

**Acknowledgments**

We would like to thank Sophie Dang, Jela Burkus, Shelby Buckley, Jenn Buss, Sara Barzczewski, Yiwen Zhang, and Yixin Huang from the University of Alberta, as well as Fanny Michaud and Sébastien Dagnault from the Canadian Forest Service – Laurentian Forestry Centre (CFS-LFC) for their help with field work and laboratory assistance.

**Funding**

The Natural Sciences and Engineering Research Council of Canada (NSERC) provided overall financial support through a
Discovery Grant (RGPIN-2014-04693) to Sylvie Quideau. This work was also funded by the Forest and Climate Change Program of the Canadian Forest Service through Jérôme Laganière and Christine Martineau. An award from the University of Alberta Northern Research and a grant from Alberta Conservation Association for Biodiversity awarded to Justine Lejoly partly funded field work and laboratory analyses.







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
