# Peer review of "Earthworm-invaded boreal forest soils harbour distinct microbial communities"

_EGUsphere, 2022_

## Author Comment (AC2)

Table A. Earthworm species found at each site after hand-sorting of the litter and hot mustard extraction for the mineral soil. Data from Lejoly et al. (2021).

| Site | Species | Functional group |
|---|---|---|
| EMEND | *Dendrobaena octaedra* Savigny | Epigeic |
| Valcartier | *Dendrobaena octaedra* Savigny | Epigeic |
| | *Aporrectodea turgida* Eisen | Endogeic |
| | *Lumbricus rubellus* Hoffmeister | Endogeic |
| | *Lumbricus* sp. | Anecic |
| Grands Jardins | *Dendrodrilus rubidus* Savigny (now *Bimastos rubidus*) | Epigeic |
| | *Aporrectodea* spp. | Endogeic |
| | *Octolasion tyrtaeum* Savigny | Endogeic |
| | *Lumbricus terrestris* L. | Anecic |

Table B. Post-hoc pairwise comparisons of the permutational ANOVA (Table 1 of the manuscript). When the interaction between site and invasion was non-significant, the differences between sites are presented. When the interaction was significant (for ITS – MIN), the comparison between invaded and control samples is presented for each site separately. Grands Jardins (GJ) is a Podzol, Valcartier (VAL) a Brunisol, and EMEND a Luvisol.

| | | Pairs | SumofSq | F model | $R^2$ | p-value |
|---|---|---|---|---|---|---|
| | | | | | | |
| Fungi | LFH | EMEND vs VAL | 1.25 | 8.7 | 0.35 | <0.01 |
| | | EMEND vs GJ | 0.95 | 6.1 | 0.33 | <0.01 |
| | | VAL vs GJ | 0.90 | 9.0 | 0.45 | <0.01 |
| | MIN | EMEND: Control vs invaded | 0.23 | 1.8 | 0.16 | <0.01 |
| | | VAL: Control vs invaded | 0.17 | 1.1 | 0.12 | 0.30 |
| | | GJ: Control vs invaded | 0.27 | 1.5 | 0.27 | 0.10 |
| Bacteria | LFH | EMEND vs VAL | 0.98 | 11.2 | 0.40 | <0.01 |
| | | EMEND vs GJ | 1.40 | 21.8 | 0.61 | <0.01 |
| | | VAL vs GJ | 0.66 | 6.8 | 0.38 | <0.01 |
| | MIN | EMEND vs VAL | 0.87 | 12.3 | 0.38 | <0.01 |
| | | EMEND vs GJ | 0.31 | 2.9 | 0.16 | <0.05 |
| | | VAL vs GJ | 0.36 | 4.2 | 0.23 | <0.01 |
| PLFA | LFH | EMEND vs VAL | 0.07 | 6.0 | 0.26 | <0.05 |
| | | EMEND vs GJ | 0.14 | 38.2 | 0.76 | <0.01 |
| | | VAL vs GJ | 0.14 | 10.6 | 0.45 | <0.01 |
| | MIN | EMEND vs VAL | 0.09 | 15.5 | 0.45 | <0.01 |
| | | EMEND vs GJ | 0.03 | 4.3 | 0.21 | <0.01 |
| | | VAL vs GJ | 0.04 | 7.5 | 0.33 | <0.01 |

Table C. Additional pairwise comparisons for the ITS – Mineral soils (where the interaction between site and invasion was significant).

| Pairs | SumofSq | F model | $R^2$ | p-value |
|---|---|---|---|---|
| EW-EMEND vs EW-VAL | 0.87 | 6.4 | 0.33 | <0.01 |
| EW-EMEND vs CONT-EMEND | 0.23 | 1.8 | 0.16 | <0.01 |
| EW-EMEND vs EW-GJ | 0.38 | 2.5 | 0.24 | <0.05 |
| EW-EMEND vs CONT-GJ | 0.46 | 3.7 | 0.32 | <0.05 |
| EW-EMEND vs CONT-VAL | 0.41 | 3.1 | 0.31 | <0.05 |
| EW-VAL vs CONT-EMEND | 0.68 | 4.7 | 0.32 | <0.01 |
| EW-VAL vs EW-GJ | 0.45 | 2.7 | 0.23 | <0.01 |
| EW-VAL vs CONT-GJ | 0.54 | 3.8 | 0.30 | <0.01 |
| EW-VAL vs CONT-VAL | 0.17 | 1.1 | 0.12 | 0.30 |
| CONT-EMEND vs EW-GJ | 0.35 | 1.9 | 0.28 | <0.05 |
| CONT-EMEND vs CONT-GJ | 0.39 | 2.9 | 0.37 | <0.05 |
| CONT-EMEND vs CONT-VAL | 0.35 | 2.3 | 0.37 | <0.10 |
| EW-GJ vs CONT-GJ | 0.27 | 1.5 | 0.27 | 0.10 |
| EW-GJ vs CONT-VAL | 0.30 | 1.4 | 0.32 | 0.20 |
| CONT-GJ vs CONT-VAL | 0.28 | 1.9 | 0.39 | 0.10 |

[Figure]

Figure A. Average (±1 SE) relative abundances (as number of reads) of fungal guilds per site. This corresponds to Figure 2 of the current manuscript.

[Figure]

Figure B. Average (±1 SE) concentrations of PLFA groups per site/soil type. This corresponds to Figure 4 of the current manuscript.

[Figure]

Figure C. Graphical abstract.

---

## Author Response (AR1)

We want to thank the two anonymous referees for their insightful and constructive feedback. We have incorporated their comments in the current manuscript, including:

- Clarifications of the sampling procedure in the methods;
- Additional figures to visualize changes in individual sites (as supplementary material);
- More information about the earthworm communities, such as a table with identified species, and the addition of one paragraph in the discussion on the differences between sites;
- Removal of unclear figures (the map and the sampling design);
- Consideration of the maple sugar site as hemiboreal;
- Updated graphical abstract;
- Additional pairwise comparisons in the PERMANOVA table

Point-by-point responses can be found below.

**Reviewer 1**

We thank the anonymous referee for their insightful comments.

Line 97: Point well taken, the sugar maple forest should be considered in the hemiboreal zone. We updated the text to include "hemiboreal and boreal forests". We also want to highlight the fact that the home range for sugar maple forests is shifting northwards with climate change, meaning that they will eventually establish themselves in regions currently covered by boreal forest types. In their recent paper, Boilard et al (2023) suggested that this shift will decrease soil carbon storage. The reference has been added.

Lines 333-341 and 355-363:

Thank you for the suggestion. Estimating the stage of invasion can be done, with caution.

For the Luvisol site (EMEND): the invasion is quite recent, with only one species (the epigeic Dendrobaenae octaedra), typically the first observed in newly invaded sites (Hale et al. 2005). The research site was established in 1999, but earthworms were first observed in 2017 (personal communication), two years before our sampling.

For the Brunisol site (Valcartier): The first mention of earthworms is recent (Lejoly et al. 2021), but earthworms were already found a few kilometers away in 2002 (Moore et al. 2009). Because of the drastic loss of forest floor (94 % of the C) and the presence of all three earthworm ecological groups (epigeic, endogeic, and anecic), it is likely that the invasion is not recent.

For the Podzol site (Grands Jardins): the first mention in the literature was made in 2004 (Moore et al. 2009) but the invasion is likely older, as the nearby lake has been used for fishing activities for more than 100 years. This suggests that this site is at a late stage of invasion, with established earthworm populations.

Both the Brunisol and the Podzol sites have earthworm biomasses and functional group composition often observed with later stages of invasion, while the Luvisol site is likely in an earlier stage.

In the current manuscript, we included a consideration of potential invasion stages for each site in the methods, based on earthworm data.

Lines 410-415:

This is a fair point. The text was updated as follows:

"Our results suggest that hemiboreal and boreal forests are affected by invasive earthworms differently from temperate forests and confirm the importance to implement addition …."

Lines 210-211: Thank you, the text has been edited.

REFERENCES:

Boilard, G., Bradley, R.L., Houle, D., 2023. A northward range shift of sugar maple (Acer saccharum) in Eastern Canada should reduce soil carbon storage, with no effect on carbon stability. Geoderma 432, 116403. https://doi.org/10.1016/j.geoderma.2023.116403

Hale, C.M., Frelich, L.E., Reich, P.B., 2005. Exotic European earthworm invasion dynamics in northern hardwood forests of Minnesota, USA. Ecological Applications 15, 848–860. https://doi.org/10.1890/03-5345

Lejoly, J., Quideau, S., Laganière, J., 2021. Invasive earthworms affect soil morphological features and carbon stocks in boreal forests. Geoderma 404, 1–13. https://doi.org/10.1016/j.geoderma.2021.115262

Moore, J.-D., Ouimet, R., Reynolds, J.W., 2009. Premières mentions de vers de terre dans trois écosystèmes forestiers du Bouclier canadien. Le naturaliste canadien 133, 31–37.

**Reviewer 2**

We would like to thank the reviewer for a thorough and constructive review of our manuscript. We will follow the reviewer's structure to clarify and answer each point. When relevant, the reviewer's text is added in italic.

1. Missing earthworms

Our reply:

We agree with the reviewer that the information regarding earthworm communities is too limited. An extensive characterization of earthworm communities was carried out in Lejoly et al (2021) and, as suggested by the reviewer, we included a table summarizing the earthworm species and functional groups found in the studied sites (now table 1).

2. Sites and sampling

Reviewer comment:

*There are some inconsistencies and unclearness around the experimental sites and the sampling procedure. In lines 97-99, four sites are mentioned - Valcartier, Grands Jardins, EMEND and Breton, - but the rest of the article, including e.g., Figure 1, Figure S1, and Table S1 lacks site Breton. I suspect Breton was not sampled then.*

Our reply:

Breton was indeed not sampled, and any mention was removed from the manuscript.

Reviewer comment:

*Another aspect connected to the applied hot mustard extraction (line 101): have soil samples been collected beforehand of this extraction? If not, could this extraction modify the soil samples' microbial communities? Or have you applied this extraction equally to control and earthworm invaded sampling points too?*

Our reply:

The earthworm survey (including the hot mustard extraction) was conducted in 2018, together with site characterization. The sampling for microbial analysis was done one year later, in 2019, at the same sites but not the same sampling point. This means that the extraction did not influence the microbial communities. We have added the year of the earthworm survey in the methods.

Reviewer comment:

*In line 103, Figure S1 is referred to explain how control and earthworm invaded patches were established per site, though this figure could not help me understand this procedure. Does Figure S1 spatially show how the pits were located, or is it just showing randomness of the pits? In general, how big the study sites were at all? Furthermore, I could not figure out the purpose of the bottom grey box in Figure S1. Also, pits are mentioned here, but it is only clear*

*from Lejoly et al. (2021), what do you mean by them. Could you just use e.g., sampling points instead, just as in line 104?*

Our reply:

Thank you for pointing this out. After consideration and based on these comments, we have decided to remove Figure S1 as it was not increasing the clarity of the manuscript. Instead, we clarified the text about the sampling protocol (e.g., lines 111-112, see below).

Reviewer comment:

*In line 110, you mention high density and low-density earthworm-invasion level for Valcartier. What is the threshold earthworm-density for these invasion stages? Is this relevant for the rest of the article, have you considered separately these invasion levels in statistical analysis?*

Our reply:

This distinction between high- and low-density earthworm invasion was indeed not relevant for the manuscript, as it was not considered in statistical analyses. This information was removed from the current version of the manuscript.

However, for clarification, this distinction was based on the intensity of changes observed for forest floor structure and thickness. In low-density zones, changes in forest floor characteristics were minimal (no significant differences in terms of thickness and C stocks compared to the control but a change from *mor* to *moder*) while in high-density zones the forest floor had almost disappeared (*mull* forest floor). For this reason, instead of mentioning the specific distinction between high and low density, we wish to include the following information: "For Valcartier, effects of earthworm invasion on the forest floor varied, ranging from minimal changes (no significant differences in terms of thickness and C stocks compared to the control but a change from *mor* to *moder*) to more drastic changes where the forest floor had almost disappeared."

Reviewer comment:

*The way of sample collection is vague in lines 111-112. Have you sampled more times the sampling points of each site through June and July? Or each sampling point was sampled once, and these samplings were spread out these two summer months? If so, and there are three to four sampling points per each site, Figure S1 is even less explanatory, where each level of invasion has exactly three sampling points/pits marked.*

Our reply:

Each sampling point was sampled once, and these samplings occurred over the months of June and July. We updated the manuscript lines 110-112 to better reflect this:

"At each site, three to six sampling points were determined for earthworm-invaded and earthworm-free zones. The sampling points were randomly distributed and sampling occurred once for each location over the months of June and July 2019."

Reviewer comment:

*Finally, Figure 1 itself does not help locating the sites, since it is unknown which red dot refers which site. In Lejoly et al. (2021), a supplementary part is present for the same figure, which shows the site locations. However, I do not see the reason to have the same map in this manuscript too. Maybe just using GPS coordinates of the sites in the text is easier and more up-to-point.*

Our reply:

Thank you for this suggestion. We have decided that it is indeed sufficient for this manuscript to only have the GPS coordinates (which are presented in Supplementary Table 1) and have therefore removed the map.

3. Result presenting and discussion

Reviewer comment:

*The outcome of this manuscript is generally interesting, however, I was wondering on the biological accurateness of presenting. If my interpretation is correct, in figures (e.g., Figure 2-4), results from all sites and all sampling points are combined. It seems strange, that these sites with differing soil type and vegetation cover (nevertheless earthworm community, see next paragraph) are discussed together, and only forest floors and mineral soils are divided. By checking Table 1, sites have a highly significant difference in their microbial communities in all studied microbial community characteristics.*

Our reply:

Instead of focusing on differences among sites/soil types, our sampling design aimed at observing general patterns at the continental scale beyond individual sites. The separate analysis of mineral soils and forest floors resulted from a significant three-way interaction (invasion*site*sample type). This was added to the section on statistical analyses as followed (line 199, end of paragraph): "Because of a significant three-way interaction, all statistical analyses were conducted separately for forest floors and mineral soils."

We also want to highlight the fact that in all cases, two-way statistical analyses were performed, meaning that site was always included as a factor.

For example, as the same patterns in terms of fungal guilds and PLFA groups of mineral soils emerged from individual sites (Figures A & B), we decided to merge them to make the figures clearer. We have decided to include site-specific figures as supplementary material.

Reviewer comment:

*I would question, whether the "interaction between invasion and site was non-significant" (lines 323-324). In lines 196-197 it is written that "the threshold for significance was set at alpha = 0.1", and by checking Table 1, there is significant interaction between invasion and site for fungi in mineral soils. After inspecting earthworm communities in Lejoly et al. (2021), it is visible, that site EMEND was rather different from the other two sites, since only one*

*epigeic species was found there, while the other sites were invaded by both epigeic, endogeic and anecic earthworms too.*

Our reply:

Point is well taken. There is indeed a significant difference that should be mentioned. From the posthoc test of the permutational ANOVA (see comment below and Table 2), all three sites harbour significantly different microbial communities, both in forest floors and mineral soils. For fungal communities of mineral soils, the difference between invaded and control samples is significant for EMEND (p-value<0.01), marginally significant for Grands Jardins (p-value=0.10), and non-significant for Valcartier (p-value=0.3). This would suggest that differences are not coming from differences in earthworm communities, but rather differences in fungal communities. Indeed, Valcartier is dominated by arbuscular mycorrhizae, and EMEND and Grands Jardins by ectomycorrhizae.

Reviewer comment:

*Even in a preceding hypothesis of any study, I would expect different outcome for earthworm effect from a 'treatment' (or a study site) with only epigeic earthworms, when compared to the influence of earthworms from all three earthworm ecological groups. Now this is only speculation without seeing a post-hoc test (which test I would highly appreciate) of the permutational ANOVA of Table 1. Even if this mentioned significant interaction difference is due to another outlining site, it would worth at least a paragraph in the discussion.*

Our reply:

We agree with the reviewer and have added the posthoc test. The three sites have significantly different microbial communities, both in forest floors and mineral soils.

For the fungal communities of mineral soils, where the interaction between site and invasion was significant, EMEND had different fungal communities between invaded and control samples (p-value < 0.01). It is still worth noticing that the differences were marginal for Grands Jardins (p-value = 0.1). The absence of differences for Valcartier aligns with the fact that the canopy is dominated by sugar maple associating with arbuscular mycorrhiza (AM) and that the ITS2 fragment cannot resolve closely related AM species (Stockinger et al, 2010). It is possible that changes are occurring in the AM fungal communities, but that we are unable to detect them.

Reviewer comment:

*Not only the sampling procedure, but also the sample size presenting was unclear. In Figure 2 and Figure 4 captions (line 247 and 314), n=7-18 and n=8-19 are present, respectively. Where these sample sizes came from? If there were indeed (at least) three sites, and each were sampled at least three times for each level of invasion, which level of invasion/soil layers had less than nine replicates?*

Our reply:

While we sampled at least three replicates per site and invasion status, we lost three samples due to mislabelling, leading to n=2 in a few instances.

We propose to clarify this at lines 196-199 with the following edits: "Threshold for significance was set at alpha =0.1 to account for the higher probability of type two error associated with the low sample size. In addition to the unfortunate loss of three samples, we recognize that regional studies such as ours necessarily have low replication. Samples (forest floor and mineral soil) were divided into two categories: invaded and non-invaded, corresponding to the factor "Invasion".

4. Other comments
4.1 On the graphical abstract:

Reviewer comment:

*The graphical abstract is a bit vague. First, the arrow pointing from the non-invaded site to the invaded one is misleading. I suppose it is intended to show the development of earthworm invaded sites, but it could also show the direction of the earthworm invasion. Also, this study is based on a "space-for-time approach" (line 19), thus it did not sample the same sampling points before and after invasion. I rather would leave the arrow out and label the two soil profiles separately.*

*I suspect, with the coloured boxes under the soil profile sketch, the most important results are presented. However, I miss clearness here. For example, do you mean fungal diversity, fungal biomass or something else with "Fungi"? If my inference is correct, then the brown box would be relative abundance of fungi, the blue is PLFA ratio and the yellow is dominant bacteria. Nevertheless, I would be more specific, that these findings refer to mineral soil, not forest floor.*

Our reply:

We thank the reviewer for requesting clarifications and for suggesting these improvements. The graphical abstract has been revised. Namely, the arrow was removed and the two soil profiles were labelled as 'control' 'and earthworm-invaded'. The box labels were also clarified, as well as the fact that these results were found for mineral soils.

4.2 Other comments

Reviewer comment:

*Lines 122-126: Is it usual and necessary to present the nucleotide sequence of the primers, when the primer code is referred?*

Our reply:

We consider that presenting the primer sequences is best for transparency while also facilitating its future use by other researchers. However, if the reviewer feels strongly about this, we are willing to remove it.

Reviewer comment:

*Line 295: Do you mean by "increase" and "decrease" a change after earthworm invasion?*

Our reply:

Thank you for pointing this out. To clarify, we mean: orange/red hues represent a significantly higher relative abundance and blue hues a significantly lower relative abundance in earthworm-invaded soils compared to controls. This is clarified in the figure caption.

Reviewer comment:

*Lines 399-400: Does not membrane fluidity usually increase with higher temperature?*

Our reply:

Unsaturation usually increases membrane fluidity, which lowers the melting point. This process is necessary to maintain homeostasis in less-than-optimal conditions and can occur with decreasing temperatures (Norris et al. 2023).

**REFERENCES**

Lejoly, J., Quideau, S., Laganière, J., 2021. Invasive earthworms affect soil morphological features and carbon stocks in boreal forests. Geoderma 404, 1–13. https://doi.org/10.1016/j.geoderma.2021.115262

Norris, C.E., Swallow, M.J.B., Liptzin, D., Cope, M., Bean, G.M., Cappellazzi, S.B., Greub, K.L.H., Rieke, E.L., Tracy, P.W., Morgan, C.L.S., Honeycutt, C.W., 2023. Use of phospholipid fatty acid analysis as phenotypic biomarkers for soil health and the influence of management practices. Applied Soil Ecology 185, 104793. https://doi.org/10.1016/j.apsoil.2022.104793

Stockinger, H., Krüger, M., Schüßler, A., 2010. DNA barcoding of arbuscular mycorrhizal fungi. New Phytologist 187, 461–474. https://doi.org/10.1111/j.1469-8137.2010.03262.x